# Characterization of a Coproduct from the Sea Cucumber *Cucumaria frondosa* and Its Effects on Visceral Adipocyte Size in Male Wistar Rats

**DOI:** 10.3390/md18110530

**Published:** 2020-10-26

**Authors:** Alan Ramalho, Nadine Leblanc, Marie-Gil Fortin, André Marette, André Tchernof, Hélène Jacques

**Affiliations:** 1School of Nutrition, Faculty of Agricultural and Food Sciences, Université Laval, Laval, QC G1V 0A6, Canada; alan.ramalho.1@ulaval.ca (A.R.); andre.tchernof@criucpq.ulaval.ca (A.T.); 2Institute of Nutrition and Functional Foods, Université Laval, Laval, QC G1V 0A6, Canada; Nadine.Leblanc@fsaa.ulaval.ca (N.L.); Andre.Marette@criucpq.ulaval.ca (A.M.); 3Fisheries and Aquaculture Innovation Centre, Merinov, Gaspé, QC G4X 2V6, Canada; Marie-Gil.Fortin@merinov.ca; 4Quebec Heart and Lung Institute, Université Laval, Laval, QC G1V 4G5, Canada; 5Department of Medicine, Faculty of Medicine, Université Laval, Laval, QC G1V 0A6, Canada

**Keywords:** sea cucumber, by-product, bioactive, saponins, cardiovascular, visceral, adipose, cardiometabolic, nutritional supplement

## Abstract

Sea cucumbers have been shown to have potential health benefits and are a rich source of several bioactive compounds, particularly triterpenoid saponins. However, most studies concentrate on the body wall, and little is known about the health effects of the coproducts. The objectives of this study were to determine the nutritional composition of a coproduct from the sea cucumber *Cucumaria frondosa* and the effects of the dietary consumption of this coproduct on cardiometabolic health in rats. Chemical, biochemical, and nutritional analyses were performed to characterize this coproduct. Forty (40) male Wistar rats were then equally divided into four groups and fed a purified control diet or a diet enriched with 0.5%, 1.5%, or 2.5% (by protein) of coproduct. After 28 days of feeding, the rats were sacrificed. Body and tissue weight, body composition, epididymal adipocyte diameter, plasma and hepatic lipids, glycemia, and insulinemia were measured at the end of the 28-day experiment. Analysis of the coproduct revealed high levels of protein, omega-3 fatty acids, minerals, and saponins. The 1.5% group had significantly smaller epididymal adipocytes vs. the control. We conclude that dietary administration of this sea cucumber coproduct at 1.5% doses decreases visceral adiposity, potentially decreasing the risk of cardiometabolic dysfunction. The coproduct’s saponin content may contribute to the observed effects, but the impact of other components cannot be ruled out.

## 1. Introduction

Cardiometabolic diseases (CMDs), including cardiovascular disease (CVD), type 2 diabetes (T2D), and non-alcoholic fatty liver disease (NAFLD) represent a worldwide major health concern [1]. For the past several decades, the global prevalence of these diseases has been increasing steadily, making CMDs the number one cause of morbidity and mortality globally [2]. An estimated 8.9 million global deaths were attributed to coronary heart disease (CHD) alone in 2015 [3]. CHD is also the main global cause of disability-adjusted life years lost [3]. The development of medications that help prevent major cardiac events (statins, fibrates, etc.) have proved quite effective [4]. However, these options are far from sufficient, due to the presence of side effects of varying gravity. Thus, there is a need for the development of effective and safer methods for the treatment and especially the prevention of CMD.

In recent years, there has been a resurgence of interest in marine bioactive compounds and their potential for the treatment of a wide range of ailments. Sea cucumbers are marine echinoderms that are of particular interest in nutrition research. They have been consumed for thousands of years in Asian countries, where they are considered a delicacy and part of traditional Asian medicine [5,6,7,8]. Scientific interest in these creatures has grown rapidly over the past few decades. Modern research has validated many potential health benefits of sea cucumbers and their extracts [8]. These benefits include anti-inflammatory, anticoagulant, antithrombotic, hypolipidemic, and hypoglycemic activities, all of which have obvious implications for the prevention and treatment of CMD [5,6,9,10,11]. Sea cucumbers are rich sources of several potent bioactive compounds, especially saponins, chondroitin sulfates, glycosaminoglycans, and sulfated polysaccharides [12]. Saponins are believed to be the cause of several of the observed health benefits due to their large range of effects [5,6,7,13]. Indeed, they have been determined to be the main bioactive compounds and the main lipid lowering component of sea cucumbers [5,13]. Previous studies indicate that the species *Cucumaria frondosa*, fished in Atlantic Canada, possesses the aforementioned potential health benefits as well as a very interesting profile of bioactive compounds [5,9,14,15]. Most research concentrates on the body wall and muscular bands of the sea cucumber, being the tissues most commonly consumed. Little research has been done specifically on the coproducts (tissues normally discarded during transformation, such as the digestive system, the respiratory tract, and the gonads). However, the few studies done on the coproducts have shown that they are very rich in saponins and other bioactive compounds, such as astaxanthin and phenols, in addition to several important nutrients such as omega-3 polyunsaturated fatty acids (PUFAs) and amino acids, particularly arginine, glycine, and glutamic acid [16,17,18,19].

Considering the interesting profile of nutrients and bioactive compounds of the coproducts of *C. frondosa*, there is potential for their use in the prevention and treatment of CMD. However, at the present time, scientific research on these products is limited. Therefore, the objectives of this study were to determine the nutritional composition of a coproduct from the sea cucumber *C. frondosa* and the in vivo effect of the dietary consumption of this sea cucumber coproduct on recognized cardio-metabolic markers and adiposity, used to evaluate cardiovascular risk, in healthy male Wistar rats. Because the coproducts contain high concentrations of various bioactive compounds with uncertain safety profiles due to a lack of studies, a preclinical trial on rats was necessary. We tested the hypothesis that dietary intake of a sea cucumber coproduct results in improvements in cardiovascular and metabolic function markers. Rats are deemed to be a representative animal model for physiological and metabolic investigations, white adipose tissue (WAT) studies, and diet–obesity studies [20]. Specifically, along with mice, rats are a common model in studies investigating the physiological effects of saponins.

## 2. Results

### 2.1. Composition and Nutrient Profile of the Sea Cucumber Coproduct

The composition and nutrient profile of the dried sea cucumber coproduct are shown in Table 1. The coproduct contained a high concentration of protein and lipids, a small amount of carbohydrates in the form of glycogen, and very little cholesterol. Regarding mineral content, the coproduct contained relatively high levels of calcium, selenium, and zinc, and very high levels of iron, potassium, sodium, and phosphorus. With respect to bioactive compounds, the coproduct contained high levels of saponins, with 178 mg/g. Previous tests performed on a similar coproduct from *C. frondosa* to measure toxic trace metals revealed that, although the concentrations of lead, cadmium, and mercury were well below acceptable limits, the concentrations of total arsenic and especially inorganic arsenic were close to the maximal acceptable limits, as set by Health Canada [21,22]. Therefore, the arsenic content of the sea cucumber coproduct used in this study was analyzed, revealing that this coproduct also contained high concentrations of total and inorganic arsenic.

Appendix A details the concentrations of the different essential and non-essential amino acids found in the coproduct. The coproduct contained high levels of essential amino acids, particularly lysine and leucine. It was also rich in many of the non-essential amino acids, notably glutamic acid, aspartic acid, glycine, alanine, and arginine. The amino acid score further demonstrates the quality of the coproduct’s protein content, with five of the essential amino acids surpassing the requirement pattern while the other four are just under the requirements.

Table 2 details the types of fatty acids contained in the coproduct’s lipids. The coproduct contained no trans-fat and little saturated fat, which represented ≈3% of the total lipid content. About one third of the lipids were in the form of phospholipids. In terms of PUFA content, the coproduct contained high concentrations of omega-3 PUFAs, mainly eicosapentaenoic acid (EPA), but very little omega-6 PUFAs. As such, omega-3 PUFAs constituted the vast majority of the total PUFAs, with a ratio of omega-3 to omega-6 of about 24 to 1. There were also 11% of the fatty acids detected in a GC/MS chromatogram that remain unidentified. They were visible in six distinct peaks (see GC/MS chromatogram in Appendix A and complete fatty acid profile in Appendix A). They were long-chain fatty acids (C ≤ 15), but their exact structure cannot be identified using Varian Saturn GC-MS software (version 6.2) and the known standards used. As a result, those fatty acids could not be assigned to their respective fatty acid classes and are presented in their own category in Table 2. The complete profile of fatty acid content of the lipids in the dried sea cucumber coproduct is presented in Appendix A.

### 2.2. Food Intake, Body and Tissue Weights, and Body Composition

As can been seen in Table 3, the initial weights of the rats were not significantly different between the groups and supplementing the diets with the coproduct did not significantly impact the average intake of food consumed per day. After 28 days, there was also no difference between the groups regarding final weight, total weight gain, total corporeal energy, total crude protein content, or total body fat mass (Table 3). However, as shown in Table 4, the average weight of epididymal adipose tissue (AT) was significantly lower in the 0.5% diet group compared with the 2.5% diet group, but not with the control group, when adjusted for covariance with initial body weight (BW) of the rats (8.36 ± 0.62 g for the 0.5% group compared to 10.7 ± 0.9 g for the 2.5% group, *p* = 0.025). No differences were observed regarding the weights of the liver, retroperitoneal and perirenal AT, kidneys, adrenal glands, extensor digitorum longus (EDL), soleus, or length of the intestines (Table 4). However, the sums of the average weights of the epididymal and retroperitoneal ATs were also significantly lower in the 0.5% group when compared with the 2.5% group (16.3 ± 1.4 vs. 20.4 ± 1.5 g, *p* = 0.021), but not when compared with the control group (Table 4). When ratios of tissue weight were calculated with final BW, the ratio of epididymal AT weight to final BW was lower in the 0.5% diet group compared to the 2.5% diet group (1.8 ± 0.1 × 10^−2^ vs. 2.2 ± 0.2 × 10^−2^, *p* = 0.017), but, again, not when compared to the control group (Table 5). The same was true for the ratio of the sum of the weights of the epididymal and retroperitoneal ATs to final BW (3.4 ± 0.2 × 10^−2^ vs. 4.1 ± 0.2 × 10^−2^, *p* = 0.012) (Table 5). The ratio of kidney weight to final BW was lower in the 2.5% group than the 1.5% group (6.4 ± 0.1 × 10^−3^ vs. 6.9 ± 0.1 × 10^−3^, *p* = 0.017), but no effect was observed when compared to the control (Table 5). The hepatic somatic index (liver weight/final weight), the ratio of EDL weight to final BW, and soleus weight to final BW did not vary between the groups (Table 5).

### 2.3. Adipocyte Diameters

As shown in Table 4, epididymal AT diameters in the 1.5% group were significantly smaller than in the control group. The diameters in the other two experimental groups were also smaller than the control group, although these differences did not reach significance.

Epididymal adipocyte size had a strong positive correlation with plasma triglycerides (TGs) (r = 0.56, *p* = 0.0003), whole body fat content (r = 0.64, *p* < 0.0001), and the relative weights of the different ATs (r = 0.71, *p* < 0.0001; r = 0.71, *p* < 0.0001; r = 0.55, *p* = 0.0003; and r = 0.54, *p* = 0.0004 for the sum of the relative weight of epididymal plus retroperitoneal fat pads, the epididymal fat pad, the retroperitoneal fat pad, and the perirenal fat pad, respectively). It also showed a moderate, positive correlation with plasma insulin (r = 0.40, *p* = 0.012), Homeostatic Model Assessment of Insulin Resistance Score (HOMA-IR) (r = 0.39, *p* = 0.017), the sum of very-low-density lipoprotein cholesterol and low-density lipoprotein cholesterol (VLDL+LDL-C) (r = 0.40, *p* = 0.014), and final BW (r = 0.40, *p* = 0.012). It additionally had significant negative correlations with crude protein content (r = −0.66, *p* < 0.0001) and the ratio of hepatic TGs to plasma TGs (r = −0.36, *p* = 0.025).

### 2.4. Plasma and Hepatic Biochemical Measurements

As shown in Table 6, a significant difference was observed in VLDL+LDL-C, where the concentration in the 1.5% diet group was significantly less than in the 0.5% diet group (0.56 ± 0.07 µg/µL vs. 0.91 ± 0.10 µg/µL; *p* = 0.046), but no effect was observed when compared with the control group. There were no significant differences between the diet groups regarding glycemia, insulinemia, plasma TGs, and high-density lipoprotein cholesterol (HDL-C). There were no differences between the groups regarding total hepatic cholesterol and TGs.

## 3. Discussion

In this study, we analyzed the nutritional content of a prepared coproduct from the sea cucumber species *C. frondosa* and evaluated its impact on recognized indicators of cardiometabolic health in healthy male Wistar rats via the administration of a purified diet supplemented with increasing amounts of this coproduct. Analysis of the coproduct revealed high contents of proteins, omega-3 PUFAs, saponins, and inorganic arsenic. The foremost result of the animal study is that, after 28 days of supplementation, the group that received 1.5% of their dietary protein from the coproduct exhibited a significant decrease in epididymal adipocyte diameters, indicating a decrease in visceral adiposity. To our knowledge, this is the first study to evaluate the impact of a sea cucumber coproduct preparation on cardiometabolic health parameters in rats, and one of the few studies to consider in vivo adipocyte size as one of the primary outcomes.

The use of sea cucumber coproducts is a distinctive aspect of this study. The coproducts that we used consisted of various tissues (mouth, thorax, tentacles, digestive tract, gonads, and respiratory tree) and were distinctly rich in saponins along with several nutrients, particularly proteins and long-chain omega-3 fatty acids. Wen et al. analyzed the chemical and nutritional composition of eight gutted and dried common commercial sea cucumber species [23]. Whereas the protein content in these eight gutted sea cucumbers (40.7% to 63.3%) is in line with the protein content of the present coproduct (43.9%), the fat content of our coproduct (32.2%) far exceeds those observed in the eight species (0.3% to 10.1%) [24]. This disparity is also seen in the study by Zhong et al., who analyzed the nutritional composition of *C. frondosa*, both with and without the internal organs [19]. This difference in lipids can be explained by the high lipid content of the internal organs. For instance, the gonads of *C. frondosa* were found to contain high levels of lipids (≈77.2 to 95.4 mg/g wet weight), particularly compared to the muscle bands (≈9.8 to 10.1 mg/g wet weight) [25]. The ratios of protein to lipids reported by Zhong et al. were far greater than that of our coproduct (7.3 to 16.68 vs. 1.36, respectively) [19]. It should be noted, however, that these ratios were much lower when internal organs were included than when they were excluded, further supporting our explanation that the coproducts have a greater lipid content than the body wall [19].

Fatty acid composition of this coproduct is also of interest. The coproduct contains significantly less saturated and monounsaturated fatty acids than the eight sea cucumber species in the study by Wen et al. [23]. The concentration of PUFAs conforms with their observations, yet the n−3/n–6 ratio was much greater in our study vs. theirs (23.8 vs. 0.25 to 0.61, respectively) [23]. Zhong et al. observed that *C. frondosa* was very rich in n−3 fatty acids, particularly EPA, which seems to be a distinction of this species [19]. Zhong et al. also reported the presence of a saturated fatty acid with a branched chain (ai-C15:0), which is unusual in most natural fats, while being predominant in bacteria, which may have a role in wound healing capabilities of sea cucumber [19]. The presence of unusual fatty acids in sea cucumber coproducts may explain why no correspondence was found between the unidentified fatty acids detected in this study and known standards of fatty acids of dietary and marine origin. Further work is needed to elucidate the exact structure of all the fatty acids of sea cucumber coproducts and to understand their effect from a nutritional standpoint.

It is also worth mentioning that the lipid composition of the viscera of *C. frondosa* varies depending on the time of the year [26]. These variations are likely due, at least in part, to changes in food availability, feeding habits, and the reproductive cycles of *C. frondosa* [26]. It has been observed that the lipid content of the viscera of *C. frondosa* fished off the coast of Nova Scotia was the highest during the winter months (January) and significantly higher than during the spring (March) and summer (July) (7.4%, 4.9%, and 5.3% of wet weight, respectively) [26]. The levels of carotenoids varied in a similar fashion, where they were significantly lower in the summer, compared to the winter and spring (1.6, 2.3, and 2.5 mg/g of lipids, respectively) [26]. The percentage of phospholipids as a function of total lipids was also significantly lower in the summer compared to the winter and spring [26]. Conversely, the percentage of PUFAs was the highest during the summer, followed by the spring, then winter (32.2%, 30.8%, and 29.6% of total lipids, respectively) [26]. EPA levels were also higher during the summer than during the spring and winter (28.2%, 25.4, and 24.7% of total lipids, respectively) [26]. Additionally, the levels of docosahexaenoic acid (DHA) were very low (0.73%, 0.84%, and 0.00% for January, March, and July, respectively) [26]. In Québec, unlike the Atlantic provinces, *C. frondosa* is mainly fished from July to October due to harsh winter weather conditions. This would effectively reduce the variability of lipid composition of the coproducts.

The saponin content is also higher in the coproduct than in the body wall. The body walls of *C. frondosa* and *Pearsonothuria graeffei* were found to contain approximately 12.1 mg/g and 100 mg/g of saponins, respectively, while our coproduct contained 178 mg/g of saponins [5,6]. This difference is supported by Bahrami et al., who also found that the relative quantity of saponins in the viscera of *H. lessoni* was higher than in the body wall [27].

The high content of inorganic arsenic was unexpected. However, it has been reported that photosynthetic organisms, such as phytoplankton and algae, are able to bioaccumulate inorganic arsenic present in their environment [28,29,30]. Benson et al., in particular, have done extensive research on this subject [31,32]. For a greater understanding of the biochemical processes involved in the accumulation and metabolism of arsenic in aquatic plants and algae, the work of Benson and his team should be consulted [31,32]. In summary, the transport system that provides these photosynthetic lifeforms with phosphate from their environment needed for their survival can also result in the absorption of environmental arsenic species, such as arsenate, which are ubiquitously present in the environment [31,32]. This absorption increases as levels of phosphate in the water decrease and approach the levels of arsenate [31,32]. In order to survive, these lifeforms have developed mechanisms that detoxify and eliminate the arsenic [31,32]. However, these mechanisms do not instantaneously convert arsenate to non-toxic forms. Benson et al. analyzed algae cultures in radioarsenate media and showed that many different arsenic species were present, including soluble inorganic species such as arsenate and arsenite [32]. *C. frondosa* are primarily benthic suspension feeders that feed on particulate matter in their proximity, namely plankton and non-living particles [16,33]. They can therefore bioaccumulate arsenic via direct exposure in the water and sediment and via the plankton they consume, leading to biomagnification of arsenic [30]. Inorganic arsenic species are highly toxic carcinogens and have been linked to CMDs such as CVD and type 2 diabetes [34,35,36]. Due to the important health concerns, regulations and recommendations on arsenic have been issued by numerous organizations. The WHO (World Health Organization) recommends the limit of arsenic in drinking water to 10 ppb [35]. The Government of Canada has established a maximum level of 3.5 ppm of arsenic in fish protein and a daily limit for adults from natural health products of <0.03 μg/kg BW/day inorganic arsenic [22,37]. The rats consumed on average 0.21 to 0.48, 0.64 to 1.46, and 1.06 to 2.49 μg/kg BW/day of inorganic arsenic in the 0.5%, 1.5%, and 2.5% groups, respectively. Thus, the amount of arsenic in the coproduct and consumed by the rats surpassed the limits as established by Health Canada for humans. However, it should be noted that humans are considerably more sensitive to arsenic than rats, and as such, we should be cautious in extrapolating the limits established for humans to rats [38].

The main result of the animal study is that at the 1.5% dosage, there was an observed decrease in visceral adipocyte size. This significant decrease in epididymal adipocyte size provides compelling evidence of the coproduct’s potential in decreasing visceral adiposity. This decrease in visceral adipocyte size was neither linked to changes in food intake nor associated with differences in body composition between groups. Therefore, we suggest that the decrease in visceral adipocyte size was not attributed to a general decrease in adiposity, but perhaps to a selective mobilization or redistribution of the lipids.

The decrease in visceral adiposity observed in the 1.5% group supports the possibility that the coproduct has the potential to benefit cardiometabolic health. Intra-abdominal or visceral fat has been linked to metabolic complications and CVDs. Increased visceral fat mass is viewed as an important independent risk factor for obesity-related metabolic dysfunction and health problems, such as atherogenic dyslipidemias, aberrant glucose–insulin homeostasis, chronic low-grade inflammation, and ectopic fat deposition, as well as being associated with increased morbidity and mortality in general [20,39,40,41,42]. In addition, as adiposity increases, so does adipocyte size in all fat deposits [41]. Adipocyte hypertrophy, particularly of visceral adipocytes, is an indicator of metabolic dysfunction in AT that is believed to independently increase the risk of CMD [40,41]. Conversely, smaller visceral adipocytes are more congruent with maintained cardiometabolic health. The validity and importance of measuring epididymal adipocyte diameter in this study is supported by its strong correlations with most of the other important cardiometabolic variables.

This decrease in visceral adipocyte size is possibly attributable to the saponin content of the coproduct. However, very few studies investigating saponins or sea cucumbers have measured adipocyte diameter. Guo et al. showed that dietary administration of an extract from *P. graeffei*, composed of 66% saponins, significantly inhibited the increase in white adipocyte size in mice caused by a high-fat diet [7]. Leal-Diaz et al. and Chen et al. also showed that plant saponins, extracted from *Agave salmiana* and *Panax ginseng*, respectively, prevented high-fat diet-induced hypertrophy in visceral adipocytes [43,44]. The agave saponins were steroidal rather than triterpenoid as they typically are in sea cucumbers [44].

As discussed, there exists a relation between the size of adipocytes and the mass of ATs. As such, the observed effects of saponins on AT mass in other studies is worthy of discussion. Several studies have reported that sea cucumber saponins can decrease the mass of ATs. Hu et al. and Meng et al. both showed that high-fat diets supplemented with saponins (0.1% and 0.08%, respectively) extracted from *P. graeffei* body walls decreased the mass of perirenal, epididymal, and subcutaneous ATs in mice [6,45]. In both studies, the masses of the ATs from the saponin groups were lower than both the high-fat and low-fat diet groups, though it was only shown to be significant for the high-fat group [6,45]. Although the rats in our study were fed a standard purified diet with a slightly elevated fat content, the decrease in AT mass that surpassed that observed with the low-fat diets in these two studies is congruent with our observations. Additionally, the doses of saponins that produced these effects (0.1% and 0.08%) are similar to the dose of saponins in the 1.5% diet in this study (≈0.12%), further increasing the validity of the claim that the saponins are responsible for the decrease [6,45]. Wang et al. also observed that supplementing a high-fat diet with either echinoside A or holothurin A, two sulfated sea cucumber saponins, decreased perirenal and epididymal AT mass in mice [46]. In another experiment, Hu et al. observed a decrease in perirenal AT weight in healthy rats fed a normal diet supplemented with 0.09 g/kg of saponins, extracted from the body walls of *C. frondosa*, when compared with the control group [5]. This dose is much smaller than any of the doses we used in our experiment, so it is probable that a higher dose would have induced similar changes in other ATs, in line with our observations. However, this may also suggest that the coproduct may be more effective at a lower dosage, which should be taken into consideration in future studies. Together, these results from other studies support the hypothesis that the saponins in the coproduct are responsible for the decrease in visceral adipocyte size.

Very little is known regarding the underlying mechanisms by which sea cucumber saponins exert these effects [47]. Several articles demonstrate that saponins have the ability to increase ß-oxidation, thermogenesis, and lipolysis, which could partly account for the decrease in visceral adiposity. Increased ß-oxidation could be caused by the activation 5’ adenosine monophosphate-activated protein kinase (AMPK), which several plant saponins have been shown to activate [48]. Chen et al. also detected that intragastric administration of ginseng saponins to high-fat diet-fed mice increased messenger ribonucleic acid (mRNA) levels of PGC-1α, as well as uncoupling protein (UCP) -1 and UCP−3 in AT, suggesting increased thermogenesis [43]. Chen et al. and Yao et al., respectively, observed a decrease in mRNA expression of peroxisome proliferator-activated receptor γ (PPARγ), adipocyte fatty acid-binding protein (aP) 2, and leptin in the epididymal AT of high-fat diet mice and a decrease in the mRNA and protein expression in adipocytes of PPARγ, CCAAT enhancer binding protein α (C/EBPα), and sterol regulatory element-binding protein 1 (SREBP-1c) in vitro, indicating that saponins inhibit fatty acid synthesis and adipogenesis [43,49]. This is further supported by the observation that quinoa saponins inhibited mRNA expression and protein levels of lipoprotein lipase (LPL), fatty acid-binding protein 4, and glucose transporter type 4 (GLUT4), as well as hindered the accumulation of TGs in adipocytes in vitro [49]. This suggests a decrease in substrate flux into adipocytes. This decrease in lipid accretion in adipocytes could additionally be accompanied by an increase in lipolysis. Liu et al. showed that adzuki bean saponins were able to increase noradrenaline-induced lipolysis in rat epididymal adipocytes in vitro [50]. Red ginseng saponins orally administered at a dosage of 0.01 g/kg BW/day for four weeks also increased the activity of post-heparin plasma LPL to normal levels in hyperlipidemic rats [51]. Considering the observation of decreased LPL mRNA and protein expression in adipocytes in vitro, the LPL expressed by the heart and skeletal muscle would be responsible for the observed increase in LPL activity [49]. This supports the concept that in the present study, the decrease in visceral adiposity is accompanied by an increase in lipid distribution to other tissues.

The coproduct’s content of omega-3 PUFAs also merits consideration. Several of the effects of omega-3 PUFAs relevant to this study are akin to the effects of saponins. Studies done in rats and mice have reported that omega-3 PUFA intake can reduce visceral AT mass and decrease the size of visceral adipocytes [52,53,54,55,56,57]. Among the possible mechanisms, dietary omega-3 PUFA administration has been shown to increase lipolysis in visceral AT, increase adiponectin, enhance AMPK activity and mRNA expression in AT, liver, and skeletal muscles, and increase lipid oxidation in WAT [52,53,55,58,59,60,61,62]. Thus, in light of these studies, we cannot exclude that the omega-3 PUFAs might have contributed to reducing adiposity. However, plasma TGs in our study did not vary significantly, which is likely due to the low concentration of omega-3 PUFA in our diets. Indeed, despite our coproduct containing high levels of omega-3 PUFAs, the actual concentration of omega-3 in the diets amounted to 0.12 mg/g, 0.35 mg/g, and 0.58 mg/g of diet in the 0.5%, 1.5%, and 2.5% diets, respectively. The lowest dosage of omega-3 used in the reviewed literature was 30 mg/g of diet, which is close to 30 times more concentrated than the 2.5% diet in our experiment [56]. Therefore, these results suggest that at these doses, the omega-3 PUFAs probably had a low impact on adiposity, which is consistent with the lack of effect on TGs.

A vast array of molecules and compounds, several of which have bioactive properties, are present in the sea cucumber coproduct. This discussion primarily focuses on the role of saponins, deemed to be the most important, but sea cucumbers are known to contain several other compounds such as collagen peptides, cerebrosides, polyphenols, antioxidants, polysaccharides, long-chain bases, and chondroitin sulfates [5,9,11,63]. Knowledge of these compounds is limited, especially those from sea cucumbers. Moreover, little is known regarding the tissular distribution of these compounds in sea cucumbers. On their own, each of these compounds could potentially affect the factors measured in this study, each with a unique effective dose and activity curve. For instance, long-chain bases from the sea cucumber *Acaudina molpadioides* were able to inhibit the size growth of epididymal adipocytes as well as the increase in epididymal, perirenal, and subcutaneous AT mass caused by a high-fat diet in mice [63]. Saponin-free dregs from *C. frondosa* were able to decrease perirenal AT weight in rats [5]. Sea cucumber cerebrosides were shown to decrease epididymal AT weight and increase expression of lipolytic and thermogenic genes [64]. Furthermore, in the present study, it is possible that when combined, these compounds have various interactions with each other, modulating their effects in an antagonistic, agonistic, or synergistic manner. Hu et al. fed rats a diet supplemented with sea cucumber body wall or purified extracts of the body wall at a dose representative of the body wall content [5]. A greater number of lipid variables were significantly reduced from the control group in the group that received the saponin extract vs. the group that received the body wall of the sea cucumber [5]. Considering that both of these groups received the same amount of saponins, this suggests that one or more constituents of the body wall countered the effects of saponins [5].

The decrease in adipocyte size observed in the 1.5% group is no longer present in the 2.5% group, which seems to align with the control. There are many possible explanations for this apparent reversal of effect. Regarding the framework of this experiment, it is possible that at a 2.5% supplementation dose, the amount of a compound reached the threshold dose needed to exert an effect, while the effectiveness of another compound had already peaked at a lower dose. Furthermore, there is the possibility that a compound has beneficial effects at the lower doses but has opposing or even toxic effects at higher doses. This could very well be the case for the saponins. Saponins are toxic to most organisms at high doses [27,65,66]. They have, depending on the molecular configuration, a high affinity to interact with cholesterol in cell membranes, create ion channels, and potentially cause lysis, killing the cell [67,68]. Hemolytic anemia and gastrointestinal lesions have been observed in rodents due to such membranolytic property [69]. As mentioned previously, the doses of coproduct used were determined based on toxicity studies of triterpene saponins, with the 2.5% group surpassing the lowest observed effect level, while the other two experimental groups contained lesser amounts of saponins. This was done, in part, to determine if sea cucumber saponins could indeed have deleterious effects at higher dosages, in this case the 2.5% group, while also having groups within the safe, non-toxic range. The use of healthy rats fed a normal fat diet rather than a high-fat diet was also done for this purpose; the contents of the coproduct, such as the saponins, are likely responsible for any detrimental changes in the variables at high doses.

The arsenic, particularly the inorganic forms that were observed in the coproduct, poses a challenge to the interpretation of our results, particularly the 2.5% group. Arsenic is a known toxin, which has been shown to detrimentally affect cardiometabolic health parameters and is believed to be a cause of T2D [70]. Indeed, arsenic can exert effects that are similar to the metabolic disturbances observed with increased visceral adiposity and metabolic syndrome, particularly insulin resistance in adipocytes [34,70,71,72]. Adebayo et al. observed an increase in WAT mass in mice exposed to 100 ppb of arsenite in drinking water, while Garciafigueroa et al. observed epididymal adipocyte hypertrophy in mice after a 5-week administration of 100 ppb arsenite in their drinking water [72,73]. On the other hand, the 2.5% rats received approximately 19.3 ppb inorganic arsenic in their diet, which was five times less than the doses administered in the studies by Adebayo et al. and Garciafigueroa et al. [72,73]. Unfortunately, none of the cardiometabolic health studies involving rodent models experimented with doses lower than 100 ppb of arsenite in the reviewed literature. Due to this lack of research, it is difficult to determine the impact the arsenic potentially had on the measured variables. However, to put this dose of arsenic into context, two different brands of grain-based laboratory rodent diet (LRD) were found to contain 19.5 to 28.6 ppb of arsenic (mainly inorganic arsenic) and 390 ppb of arsenic (56 ppb of inorganic arsenic) [74,75]. Additionally, there is evidence that saponins and selenium are both capable of countering toxic effects of arsenic in the liver [76,77]. Manna et al. postulated that some saponins may chelate with arsenic and thereby inhibit its effects [76]. Thus, it is not possible to ascertain the effect the arsenic may have had on the results, and therefore we cannot refute that the arsenic may have contributed to the disappearance of effects in the 2.5% group.

Although no effect was observed in plasma lipids, the serum lipoproteins, as well as in hepatic lipids in relation to the control group, the VLDL+LDL-C had a significant positive correlation with the plasma TGs (r = 0.66, *p* < 0.0001), which in turn had a significant positive correlation with epididymal adipocyte diameters (r = 0.40, *p* = 0.014). These correlations indirectly suggest the existence of a potential metabolic cascade, starting from the adipocyte size. The nonsignificant decrease in VLDL+LDL-C in the 1.5% group could indeed be a result of the nonsignificant decrease in plasma TG, which in turn could be a result of the decrease in adipocyte size.

One strength of this study is the use of purified ingredients to formulate diets that in turn were isoenergetic, isolipidic, isonitrogenous, and respected the AIN-93A guidelines for providing the rats with all the necessary nutrients [78]. Using purified ingredients eliminates the variability in nutrient content inherent with a chow diet, assuring that observed differences between groups are solely attributable to the addition of sea cucumber coproduct.

We acknowledge that the rats were healthy and fed a standard diet with only a slightly elevated fat content. It is possible that the impact of the coproduct on the measured markers would have been exacerbated with a high-fat or high-sucrose diet.

Another limitation is the lack of speciation of the bioactive molecules. Total quantities of saponins and inorganic arsenic were determined, but the active compounds remain unknown. Sea cucumbers are known to contain many other types of bioactive compounds, such as cerebrosides, that were not measured in this study.

Regarding the arsenic, it would be prudent to attempt to decontaminate the coproduct in future studies via methods such as membrane filtration, activated carbon filtration, or adsorption on zeolites.

## 4. Materials and Methods

### 4.1. Sea Cucumber Coproduct

The sea cucumber coproduct (SCC) was supplied by Merinov (Gaspé, QC, Canada). Merinov collected fresh sea cucumber by-products directly from the processing line at Poissonnerie de Cloridorme Inc. (Cloridorme, QC, Canada). All by-products, including the mouth, thorax, tentacles, digestive tract, gonads, and respiratory tree, were immediately transported to the Merinov fractionation center where they were blended and ground with a commercial grinder (Urschel^®^ Comitrol^®^ Processor, model 3600, Urschel Laboratories, Inc., Chesterton, IN, USA), homogenized, and then frozen at −30 °C. The by-products were then dried at 40 °C in a cooking cell (Brute-Vaillant model BOV-SM-1) for 24 h. The dried sea cucumber coproduct was vacuum sealed to prevent oxidation and kept frozen until preparation of the diets. The resulting powdered coproduct was granular and somewhat heterogenous but was fine enough to be incorporated into the purified diets without further transformation.

#### 4.1.1. Determination of the Chemical Composition of the Sea Cucumber Coproduct

The sea cucumber coproducts underwent analysis to determine their chemical composition. Moisture and total minerals (ash) were measured using the official analytic methods of the Association of Official Analytical Chemists (No. 950.46 and No. 938.08) [79]. Lipids were analyzed using a modified Bligh and Dyer method [80]. Crude proteins were determined by the Kjeldahl method (nitrogen X 6.25) adapted from the official method of AOAC No. 988.05 [79]. Glycogen was determined using the anthrone reagent method of De Zwaan and Zandee [81].

#### 4.1.2. Analysis of the Cholesterol, Saponin, and Carotenoid Content of the Sea Cucumber Coproduct

Cholesterol was quantified using a commercial ELISA kit (Cholesterol/Cholesteryl Ester Quantitation Kit, KA0829, Abnova, Walnut, CA, USA) according to the manufacturer’s protocol. Saponins were extracted with methanol and quantified by spectrophotometry according to the method of Ncube et al. [82,83]. Carotenoids were extracted with cold acetone and transferred to petroleum ether. Carotenoid concentration was determined by measuring the absorbance at 468 nm [84].

#### 4.1.3. Lipid Composition and Quality of the Sea Cucumber Coproduct

Total phospholipids were quantified by gravimetry with Solid Phase Extraction (SPE) DSC-diol columns. Neutral lipids were eluted with chloroform, and phospholipids were extracted with methanol with 0.5 mL/100 mL of 25% ammoniac solution [85]. The anisidine value (p-AnV) was determined according to official method Cd18-90 [86]. The peroxide value (PV) was determined according to method 965.33 [86]. Total oxidative value was calculated according to the formula 2PV + p-AnV [87]. Fatty acid composition of lipids in the sea cucumber coproduct was determined by transesterification of lipids in fatty acid methyl esters (FAMES). These FAMES were then separated and quantified by gas chromatography with a Varian^®^ column VF−23 ms (90 m × 0.25 mm, ID 0.25) (Varian Inc., Palo Alto, CA, USA) with flame ionization detection as previously described [88]. All analyses were performed in duplicate.

#### 4.1.4. Amino Acid Analysis of the Sea Cucumber Coproduct

For the determination of amino acids, an acidic hydrolysis was performed with 6 N hydrochloric acid at 110 °C for 23 h [89]. The reagent 6-aminoquinolyl-N-hydroxy-succinimidyl carbamate (AccQ-Tag Ultra, Waters Corporation, Milford, MA, USA) was used for derivatization of amino acids. The chromatographic separation of amino acids was performed with a column (Waters AccQ-Tag Ultra Acquity UPLC BEH C18 (2.1 mm i.d. × 100 mm × 1.7 µm particle size)) installed on a high-performance liquid chromatography system (Acquity UPLC^®^ H-Class) composed of a quaternary solvent manager pump (QSM), a refrigerated sample manager flow-through needle (SM-FTN), a column oven and a photodiode array detector (PDA). A quaternary gradient was applied during separation of amino acids, as described in Szkudzińska et al. [90]. Tryptophan analysis was performed separately with high-performance liquid chromatography, following the method of Sanchez-Machado et al. [91]. In summary, basic hydrolysis of protein hydrolysates was performed in 4.2 M sodium hydroxide for 4 h at 120 °C. Then, the pH was adjusted to 9 with concentrated hydrochloric acid. The excitation wavelength was set at 280 nm and the emission at 348 nm. The column used was an Inertsil ODS−4150 mm × 4.6 mm, 5-micron particles (GL Sciences, Tokyo, Japan, provided by Canadian Life Science, Ontario, Canada). The isocratic elution system consisted of a mobile phase of 40 mM sodium acetate:methanol 80:20 (v/v) and a flow rate of 0.8 mL/minute.

The amino acid scores were calculated according to the method outlined by the Food and Agriculture Organization of the United Nations, the WHO, and United Nations University (FAO/WHO/UNU) using the adult indispensable amino acid requirements as determined by the FAO/WHO/UNU [92]. To calculate the scores, the amino acid content of the coproduct (mg amino acid/g protein) was divided by the requirement pattern of the same amino acid (mg amino acid/g protein) [92].

#### 4.1.5. Mineral Analysis of the Sea Cucumber Coproduct

Analysis of calcium, iron, potassium, sodium, phosphorus, selenium, and zinc in sea cucumber extract was performed by Groupe Environnex (Longueuil, QC, Canada) using microwave digestion and inductively coupled plasma–mass spectrometry (ICP-MS) [93]. Analysis of arsenic, total organic arsenic, and total inorganic arsenic in sea cucumber by-products was performed by SGS Canada Inc. (Burnaby, BC, Canada) using ICP-MS and standard methodologies certified by the Canadian Association for Laboratory Accreditation.

### 4.2. Diets

Table 7 presents the formulation of the four study diets: one control diet (casein) and three experimental diets supplemented with incrementally increasing concentrations of SCC (0.5%, 1.5%, 2.5% of total protein content). These concentrations were selected based on the coproduct’s saponin content (178 mg/g), the lowest no-effect level (50 mg/kg/day), and the lowest observed effect level (150 mg/kg/day) for plant-derived triterpene saponins in male rats from toxicity studies [69,94]. The 0.5%, 1.5%, and 2.5% concentrations, respectively, represent saponin concentrations at the no-effect level (47.9 mg/kg/day), slightly below (143.7 mg/kg/day) the lowest observed effect level, and largely above (239.5 mg/kg/day) the lowest observed effect level, when considering the expected initial weight of the rats. The diets were formulated to be isoenergetic, isolipidic, isonitrogenous, and to provide all the nutrients needed to promote proper growth and health in rats, in accordance with the American Institute of Nutrition AIN-93G rodent diet, [78]. To achieve this, the amounts of casein, lard, and starch added to the diets were adjusted accordingly in order to keep the ratios of macronutrients constant between the four diets (protein:lipids:carbohydrates 20 g:15 g:52.5 g). The total energy content of each diet was determined by means of calorimetric bomb with an automatic adiabatic calorimeter (Model 1241, Parr instrument Co., Moline, IL, USA). The protein content of the casein used was measured by combustion using a nitrogen/protein analyzer (LECO FP-528, LECO Inc., St. Joseph, MI, USA).

### 4.3. Animals and Experimental Design

Forty male albino Wistar rats were purchased from Charles River Laboratories (Saint-Constant, QC, Canada). The rats were housed individually in solid-bottom plastic cages at Université Laval’s campus animal facility (Université Laval, QC, Canada). Atmospheric conditions were controlled with an ambient temperature of 20 ± 2 °C, a humidity of 45 to 55%, and a light/darkness cycle of 12 h/12 h. Food and tap water were supplied ad libitum during the entire protocol. After an acclimation period of 7 days, during which they were fed rodent chow (Purina Canada, Mississauga, ON, Canada), the rats were weighed and randomly assigned to one of the four diet groups (10 rats/diet). At the time of randomization, the rats weighed approximately 200 g.

All procedures were approved by Université Laval’s Institutional Animal Care Committee in accordance with the guidelines of the Canadian Council on Animal Care (file #2017-031-1). The animal protocol began with a seven-day adaptation period that allowed the rats to gradually become accustomed to the experimental diets. The ratio of protocol diet to ground rodent chow was 25:75 for three days, then 50:50 and 75:25 for two days each. During this period, the BW of the rats and the amount of food ingested were weighed and recorded daily. Thereafter, the rats were fed 100% their respective experimental or control diets for 28 days. Then, the BW of the rats and the amount of food consumed were recorded every two days.

### 4.4. Data Collection

#### 4.4.1. Tissue Collection

On day 29, after a 12 h fast, the rats were anesthetized with 3% isoflurane. Approximately 10 mL of blood was collected via aortic puncture. The samples were divided into blood collection tubes for plasma (two BD Vacutainer^®^ K2 EDTA 10.8 mg 6 mL tubes, BD, Franklin Lakes, NJ, USA) and for serum (one Vacuette^®^ 2 mL Z Serum Clot Activation Tube, Greiner Bio-One, Kremsmünster, Austria). The plasma tubes were immediately placed on ice. The serum tubes were allowed to clot for a minimum of 30 min at room temperature. The rats were then sacrificed via exsanguination.

In order to isolate serum and plasma, blood samples were centrifuged at 2500 rpm for 10 min at room temperature immediately after sacrifice. The serum and plasma were then aliquoted into labeled microcentrifuge tubes, flash frozen in liquid nitrogen, and stored at −80 °C until further measurements.

Once sacrificed, the liver, the epididymal, perirenal, and retroperitoneal ATs, the kidneys, the adrenal glands, the hind right EDL muscle, and the hind right soleus muscle were all extracted and weighed. The intestines were also extracted, emptied of their contents, and their length was measured. The liver was excised, the four lobes were separated, flash frozen in liquid nitrogen, and stored at −80 °C for future analyses. Half of each of the ATs were separated into three aliquots of equal size, flash frozen in liquid nitrogen, and stored at −80 °C for future analyses. All the other extracted tissues were returned to the rat’s body for later analysis of body composition. The carcasses were stored at −20 °C until they were processed for further analyses.

The three visceral fat pads measured in this study were chosen mainly due to their frequent use in the literature as well as their accessibility in rats [95]. Although the mesenteric fat pad may be deemed more comparable to human visceral AT, it was not examined here because it is difficult to isolate from the surrounding tissue and was not considered in previous studies [39].

#### 4.4.2. Body Composition

Body composition was determined using the method described by Jacques et al. [96]. Crude protein content was determined by the AOAC official method 2001-11 (Foss Kjeltec 8400 analyzer unit; Foss Co., Hillerod, Denmark). Total energy content was determined via calorimetric bomb with an automatic adiabatic calorimeter (Model 1241, Parr Instrument Co. Moline, IL, USA). Energy from lipids was determined to be the difference between the total energy and the energy from the crude protein (23.51 kJ/g), with the assumption that the carbohydrate content was insignificant. The energy from lipids was then used to determine the lipid content of each carcass (39.29 kJ/g).

#### 4.4.3. Hepatic Lipid Extraction and Measurement

Total hepatic lipids were extracted from liver samples using a modified Folch method [97,98]. Briefly, liver samples were ground in 0.9 mL of 2:1 chloroform:methanol solution with a high torque overhead stirrer. Then 0.3 mL of chloroform were added to the tube and left to agitate for 2 h. After centrifugation for 15 min at 4000 rpm at 4 °C, 0.825 mL of the supernatant were mixed with 0.4 mL of chloroform and 0.275 mL of NaCl 0.73%. After another three minutes of centrifugation at 4000 rpm at 4 °C, the superior phase was discarded, and the inferior phase underwent three rinses with 0.8 mL of Folch solution (three parts chloroform:48 parts methanol:47 parts 0.58% NaCl). A few drops of methanol were added until the remaining mixture became transparent. The contents of the tubes were then evaporated under nitrogen and stored at −20 °C until analysis.

Total hepatic cholesterol was determined using an enzymatic kit from Randox Laboratories (RDX-CH200, Randox Laboratories, Crumlin, UK) [99]. One hundred milliliters of isopropanol + 10% Triton X-100 were added to each tube to dissolve the lipid residue. Due to the high concentration of cholesterol of the samples, they underwent a 10-fold dilution with pure isopropanol. Two hundred microliters of Randox^®^ Total Cholesterol Reagent was added to the wells of the assay plate, followed by 2 µL of the controls and the samples. After an incubation period of five minutes at 37 °C, the absorbance was measured at 500 nm.

A further 10-fold dilution of the samples was required for the hepatic TG assay. This dilution was performed by adding 20 µL of the samples to 180 µL of pure isopropanol. TG concentrations were enzymatically determined using the Infinity™ Triglyceride Reagent Kit (TR22421, Thermo Scientific, Waltham, MA, USA) [98]. Two hundred microliters of the Infinity™ Triglycerides Liquid Stable Reagent were added to each of the wells followed by 2 µL of the controls and the samples. After an incubation period of five minutes at 37 °C, absorbance was measured at 500 nm.

#### 4.4.4. Plasma and Serum Analyses

##### Fasting HDL Cholesterol, VLDL+LDL Cholesterol, and Triglycerides

Colorimetric assay kits were used to assess serum levels of high-density lipoprotein (HDL) cholesterol and very-low-density lipoprotein (VLDL) + low-density lipoprotein (LDL) cholesterol (Ab65390, Abcam, Cambridge, MA, USA). Plasma TGs were measured with a colorimetric assay kit that uses enzymatic hydrolysis to separate fatty acids from glycerol to then measure the free glycerol with a coupled enzymatic system (Item No. 10010303, Cayman Chemical, Ann Arbor, MI, USA).

##### Glucose–Insulin Homeostasis

Fasting blood glycemia was measured at the time of sacrifice using aortic blood and a Bayer Contour^®^ blood glucose monitor and strips. An ELISA assay kit and a diabetes antigen control (rat/mouse) were used to measure plasma insulin (Catalog No. 10-1250-01 and 10-1220-01, Mercodia AB, Uppsala, Sweden). The calculation of the HOMA-IR score was done using the following formula:(1)$$\mathrm{HOMA}-\mathrm{IR}=\frac{\mathrm{Insulin}\;\left(\frac{\mathrm{IU}}{L}\right)*\mathrm{Glycemia}\left(\frac{\mathrm{mmol}}{L}\right)}{22.5}$$

The conversion of 1 IU of rat insulin being equal to 7.174 pmol was used [100].

#### 4.4.5. Histopathologic Assessment

Histological samples of the left lateral lobe of the rat livers and the rat epididymal ATs were prepared from the collected samples stored at −80 °C according to the method described by Laforest et al. [101]. The samples were prepared by separating each of the frozen tissues on dry ice into 100 to 600 mg pieces. The surgical equipment was kept chilled with dry ice and liquid nitrogen. The tissues were frequently submerged in liquid nitrogen to prevent thawing. Once a tissue sample was weighed, it was immediately placed in a histology cassette, submerged in 10% phosphate-buffered formalin, pH 7.0, chilled to ≈4 °C, and kept on ice until all the tissue samples were prepared. The samples were then stored at 4 °C for 24 to 48 h until they were transported on ice to be embedded in paraffin at the research center of the Centre Hospitalier de l’Université Laval (CRCHUL, Québec, QC, Canada). Once embedded, slides were prepared and underwent hematoxylin and eosin (H&E) staining.

Microscopic images were taken at 20× magnification using a Zeiss Axiocam 506 mono attached to a Zeiss Axio Observer Z1 microscope (Carl Zeiss, Oberkochen, Germany), located at the Quebec Heart and Lung Institute (IUCPQ, Quebec, QC, Canada).

Adipocyte size was determined using ImageJ software (version 1.52q) and the method described by Laforest et al. [101]. Briefly, five subregions of the original image for each rat epididymal AT slice were randomly selected. Using the Zeiss Zen pro software (Zen 2 version 2.0.0.0), these images were then denoised and converted into TIF format. Examples of representative subregions of each diet group are shown in Appendix A. In ImageJ, the images were converted to 8 bit, the background was subtracted, the threshold was adjusted for optimal contrast, and then the image was converted to binary. Examples of representative subregions of each diet group that have been converted are shown in Appendix A. Once this was done, each snap was analyzed in a predetermined random order to identify the cells with a complete membrane delimiting a single lipid droplet. The areas of these cells were then measured in pixels2, the diameters in pixels were calculated, and then the diameters were converted from pixels to µm using a ratio of 0.212. In cases were image quality was not optimal, the adipocyte membrane was filled in using the paintbrush tool and the flood tool was used to remove noise in the cell. At least 100 cells were measured for each rat. In the cases where there were fewer than 100 measurable cells in the images, these rats were omitted from the analysis.

### 4.5. Statistical Analysis

Statistical analysis was performed using SAS Studio 3.8 University Edition (SAS Institute Inc., Cary, NC, USA). All measurements, tests, and assays were done in duplicate, unless specified. The measurements of HDL-C and VLDL+LDL-C were analyzed in singlicate. The analyses of fatty acid content and amino acids were conducted in triplicate and in four replicates, respectively. All data are expressed as means ± standard error of the mean (SEM), except where specified. One-way analysis of variance (ANOVA) using the generalized linear model (GLM) procedure was used to compare the four diet groups for the initial weight of the rats. Normality and homogeneity of variance was tested using the Shapiro–Wilk test and Levene’s test, respectively. When a significant difference was detected, Tukey’s honest significant difference (HSD) test was used to determine between group differences for normally distributed and non-normally distributed variables, respectively.

Using the GLM procedure, an analysis of covariance (ANCOVA) with the initial corporeal weight of the rats as the covariate was performed to compare the four diet groups for the remaining variables. Inter-group differences were determined using the least square means (lsmeans) test and the Tukey adjustment. Normality was tested using the Shapiro–Wilk test and homogeneity of variance was verified using residual plots. Due to not being normally distributed, liver weight, kidney weight, liver somatic index, kidney somatic index, insulinemia, HOMA-IR, plasma TG levels, VLDL+LDL-C, and total hepatic TG concentration were log 10 transformed and the length of the intestines underwent a square transformation. No further transformations were required to obtain a normal distribution. Due to heterogeneity of variances, retroperitoneal AT weight and the ratio of EDL weight to final BW underwent cubic and arcsine transformations, respectively.

The mixed model (MIXED) procedure was used to perform an analysis of covariance (ANCOVA) with epididymal adipocyte diameters. To detect associations between variables, Pearson correlation coefficients were calculated using the Pearson correlation (CORR) procedure. Average epididymal adipocyte diameter was used for each rat for the calculation of the correlation coefficients.

To simplify understanding of the tables, when a significant difference exists between at least one experimental group and the control, an asterisk is placed by the *p*-value. Significance was set at *p* ≤ 0.05.

## 5. Conclusions

In conclusion, this study has shown that dietary supplementation in healthy Wistar rats with a coproduct from the sea cucumber *C. frondosa* resulted in a decrease in visceral adiposity, as evidenced by a decrease in visceral adipocyte size at the dose of 1.5%. This supports our initial hypothesis that the coproduct can benefit cardiometabolic health. This decrease disappeared in the 2.5% group, suggesting that higher doses of the coproduct do not benefit cardiometabolic health. The decrease is potentially attributed to the presence of saponins contained within the coproduct since saponins are believed to be the principal bioactive compound in sea cucumbers in both concentration and biological effects. However, sea cucumbers contain other compounds that could also impact cardiometabolic health and thus should not be ignored. Further studies are necessary to elucidate the true impact and optimal safe dosage of the coproduct on cardiometabolic health.

## Figures and Tables

**Table 1 marinedrugs-18-00530-t001:** Chemical, biochemical, and macronutrient profile of the dried sea cucumber coproduct (mean ± standard deviation).

	Parameter	Concentration
Macronutrients	Total protein (g/100 g DW ^1^)	43.9 ± 0.2
	Glycogen (g/100 g DW ^1^)	6.2 ± 0.8
	Total lipids (g/100 g DW ^1^)	32.2 ± 0.4
	Cholesterol (µg/g DW ^1^)	0.43 ± 0.04
Minerals (µg/g DW ^1^)	Calcium	1350 ± 71
	Iron	88 ± 3
	Potassium	20,000 ± 0
	Sodium	19,000 ± 0
	Phosphorus	11,000 ± 0
	Selenium	2.25 ± 0.07
	Zinc	63.5 ± 3.5
Bioactive compounds	Carotenoids (µg/g DW ^1^)	636.1 ± 24.3
	Saponins (mg/g DW ^1^)	178 ± 17
Arsenic (µg/g DW ^1^)	Organic	5.90
	Inorganic	1.69
Other	Humidity (g/100 g DW ^1^)	7.68 ± 0.10
	Peroxide value (PZ) (mEq/kg humid product)	4.39 ± 0.64
	P-anisidine value (p-AnV)	30.3 ± 3.5
	Total oxidative value (2 PV + p-AnV)	39.1 ± 2.3

^1^ DW = dry weight.

**Table 2 marinedrugs-18-00530-t002:** Fatty acid content of the lipids in the dried sea cucumber coproduct (mean ± standard deviation; *n* = 3).

Lipid Class	Concentration (mg per g of Oil)	Percentage (%)
Neutral lipids	576 ± 19	57.6 ± 1.9
Phospholipids	328 ± 13	32.8 ± 1.3
Saturated fatty acids	31.9 ± 1.3	3.19 ± 0.13
Trans fatty acids	0	0
Mono-unsaturated fatty acids	113 ± 3	11.3 ± 0.3
Poly-unsaturated fatty acids		
Sum of omega-3 and -6	164 ± 6	16.4 ± 0.6
Omega−3	157 ± 5	15.7 ± 0.5
Omega-6	6.60 ± 1.02	0.660 ± 0.102
Unidentified fatty acids	111 ± 7	11.1 ± 0.7

**Table 3 marinedrugs-18-00530-t003:** Dietary intake, body weight, and body composition of the rats (g) (mean ± standard error).

	Diet ^1^				
	Control	0.5	1.5	2.5	*p* ^2^
Intake	26.1 ± 0.6	25.6 ± 0.7	25.9 ± 0.6	26.4 ± 0.7	0.794
Initial weight	205 ± 2	204 ± 4	205 ± 1	205 ± 4	0.998 ^3^
Final weight	485 ± 10	471 ± 14	480 ± 7	489 ± 13	0.483
Total weight gain	280 ± 9	267 ± 10	275 ± 7	284 ± 10	0.483
Crude protein content (g/100 g dry weight)	51.6 ± 1.4	52.7 ± 1.6	52.3 ± 1.2	50.7 ± 1.6	0.741
Fat mass (g/100 g dry weight)	37.3 ± 1.6	36.3 ± 1.9	36.7 ± 1.5	38.7 ± 1.7	0.711
Total corporeal energy (Cal/g)	6410 ± 73	6370 ± 90	6390 ± 70	6490 ± 70	0.693

^1^*n* = 10/diet. ^2^ Adjusted for initial weight as a covariate. ^3^
*p*-value obtained from an unadjusted one-way ANOVA.

**Table 4 marinedrugs-18-00530-t004:** Tissue measurements (mean ± standard error).

Tissue	Diet ^1^				*p* ^2^
	Control	0.5	1.5	2.5	
Liver (g)	23.6 ± 0.9	23.8 ± 1.2	23.9 ± 0.5	23.9 ± 1.2	0.993 ^3^
Epididymal AT (g)	10.1 ± 0.6 ^ab^	8.4 ± 0.6 ^a^	8.7 ± 0.6 ^ab^	10.7 ± 0.9 ^b^	0.025
Retroperitoneal AT (g)	9.3 ± 0.6	7.9 ± 0.9	8.0 ± 0.5	9.6 ± 0.7	0.069 ^4^
Epididymal + Retroperitoneal AT (g)	19.4 ± 1.2 ^ab^	16.3 ± 1.4 ^a^	16.8 ± 0.9 ^ab^	20.4 ± 1.5 ^b^	0.021
Perirenal AT (g)	3.3 ± 0.4	2.9 ± 0.3	3.2 ± 0.4	3.7 ± 0.4	0.390
Kidneys (g)	3.2 ± 0.1	3.1 ± 0.1	3.3 ± 0.1	3.1 ± 0.1	0.172 ^3^
Suprarenal glands (g)	0.10 ± 0.01	0.10 ± 0.01	0.10 ± 0.01	0.11 ± 0.01	0.950
EDL (g)	0.84 ± 0.02	0.85 ± 0.04	0.92 ± 0.04	0.87 ± 0.04	0.431
Soleus (g)	0.21 ± 0.01	0.21 ± 0.01	0.21 ± 0.01	0.23 ± 0.02	0.542
Intestines (cm)	138 ± 2	130 ± 3	132 ± 3	135 ± 4	0.287 ^5^
Epididymal adipocyte diameters (µm)	69.0 ± 1.9 ^b^	63.9 ± 1.8 ^ab^	61.2 ± 1.8 ^a^	65.4 ± 1.9 ^ab^	0.032 *

^1^*n* = 10/diet. ^2^ Adjusted for initial weight as a covariate. ^3^ The *p*-value was obtained after a base 10 logarithmic transformation to produce a normal distribution. ^4^ The *p*-value was obtained after an exponential transformation to the power of 3 to produce a homogeneous variance. ^5^ The *p*-value was obtained after an exponential transformation to the power of 2 to produce a normal distribution. * At least one experimental group is significantly different from the control group. ^ab^ Values with the same exponent are not significantly different (*p* > 0.05). AT = Adipose tissue. EDL = Extensor digitorum longus.

**Table 5 marinedrugs-18-00530-t005:** Ratio of tissue weight vs. final body weight (g/g) (mean ± standard error).

	Diet ^1^				
	Control	0.5	1.5	2.5	*p* ^2^
Liver/final weight	4.9 ± 0.1 × 10^−2^	5.0 ± 0.2 × 10^−2^	5.0 ± 0.0 × 10^−2^	4.9 ± 0.1 × 10^−2^	0.688 ^3^
Epididymal AT/final weight	2.1 ± 0.1 × 10^−2 ab^	1.8 ± 0.1 × 10^−2 a^	1.8 ± 0.1 × 10^−2 ab^	2.2 ± 0.2 × 10^−2 b^	0.017
Retroperitoneal AT/final weight	1.9 ± 0.1 × 10^−2^	1.7 ± 0.2 × 10^−2^	1.7 ± 0.1 × 10^−2^	2.0 ± 0.1 × 10^−2^	0.128 ^3^
Epididymal + Retroperitoneal AT/final weight	4.0 ± 0.2 × 10^−2 ab^	3.4 ± 0.2 × 10^−2 a^	3.5 ± 0.2 × 10^−2 ab^	4.1 ± 0.2 × 10^−2 b^	0.012
Perirenal AT/final weight	6.7 ± 0.6 × 10^−3^	6.0 ± 0.4 × 10^−3^	6.5 ± 0.6 × 10^−3^	7.6 ± 0.8 × 10^−3^	0.429
Kidneys/final weight	6.5 ± 0.2 × 10^−3 ab^	6.5 ± 0.1 × 10^−3 ab^	6.9 ± 0.1 × 10^−3 b^	6.4 ± 0.1 × 10^−3 a^	0.043
EDL/final weight	1.7 ± 0.1 × 10^−3^	1.8 ± 0.1 × 10^−3^	1.9 ± 0.1 × 10^−3^	1.8 ± 0.1 × 10^−3^	0.412 ^4^
Soleus/final weight	4.3 ± 0.2 × 10^−4^	4.4 ± 0.2 × 10^−4^	4.3 ± 0.2 × 10^−4^	4.7 ± 0.4 × 10^−4^	0.686

^1^*n* = 10/diet. ^2^ Adjusted for initial weight as a covariate. ^3^ The *p*-value was obtained after a logarithmic base 10 transformation to produce a normal distribution. ^4^ The *p*-value was obtained after an arcsine transformation to produce a homogeneous variance. ^ab^ Values with the same exponent are not significantly different (*p* > 0.05).

**Table 6 marinedrugs-18-00530-t006:** Plasma and hepatic biochemical measurements (mean ± standard error).

	Diet ^1^				
	Control	0.5	1.5	2.5	*p* ^2^
Glycemia (mmol/L)	9.3 ± 0.5	10.9 ± 0.5	10.3 ± 0.3	10.2 ± 0.5	0.136
Insulinemia (µg/L)	0.56 ± 0.10	0.60 ± 0.12	0.72 ± 0.14	1.29 ± 0.45	0.468 ^3^
Glycemia/insulinemia ratio (mmol/L/pmol/L)	0.12 ± 0.02	0.15 ± 0.03	0.11 ± 0.02	0.12 ± 0.04	0.667 ^3^
HOMA-IR	5.61 ± 1.01	6.94 ± 1.50	7.81 ± 1.40	14.6 ± 5.5	0.358 ^3^
Plasma triglycerides (mg/dL)	77.3 ± 14.0	77.6 ± 5.1	62.5 ± 4.0	75.1 ± 11.4	0.597 ^3^
HDL-C (µg/µL)	0.43 ± 0.04	0.43 ± 0.05	0.33 ± 0.03	0.44 ± 0.04	0.145
VLDL + LDL-C (µg/µL)	0.69 ± 0.09 ^ab^	0.91 ± 0.10 ^b^	0.56 ± 0.07 ^a^	0.68 ± 0.08 ^ab^	0.046 ^3^
Total hepatic cholesterol (µg/mg)	26.0 ± 1.6	28.3 ± 1.6	29.9 ± 2.0	26.3 ± 1.8	0.373
Total hepatic triglycerides (µg/mg)	79.5 ± 5.9	81.9 ± 5.1	99.7 ± 11.5	80.3 ± 5.6	0.267 ^3^

^1^*n* = 10/diet. ^2^ Adjusted for initial weight. ^3^ The *p*-value was obtained after a base 10 logarithmic transformation to produce a normal distribution. ^ab^ Values with the same exponent are not significantly different (*p* > 0.05). HOMA-IR = Homeostatic Model Assessment of Insulin Resistance. HDL-C = High-density lipoprotein cholesterol. VLDL + LDL-C = The sum of very-low-density lipoprotein cholesterol and low-density lipoprotein cholesterol.

**Table 7 marinedrugs-18-00530-t007:** Formulation of the experimental diets (per 100 g of diet).

Component	Diet			
	Control	0.5	1.5	2.5
SCC ^1^ (g)	0	0.228	0.684	1.140
Casein ^2^ (g)	22.47	22.36	22.13	21.91
Lard (g)	10.0	9.93	9.78	9.63
Soybean oil (g)	4.00	4.00	4.00	4.00
Starch (g)	32.53	32.49	32.40	32.32
Sucrose (g)	20	20	20	20
Cellulose (g)	5	5	5	5
Cholesterol (g)	1	1	1	1
Minerals ^3^ (g)	3.5	3.5	3.5	3.5
Vitamins ^4^ (g)	1	1	1	1
Choline bitartrate (g)	0.3	0.3	0.3	0.3
BHT (g) ^5^	0.2	0.2	0.2	0.2
Energy (Cal) ^6^	4852 ± 6	4862 ± 3	4883 ± 16	4897 ± 12

^1^ SCC = Sea cucumber coproduct. ^2^ Casein was determined to be 89.01% protein. ^3^ AIN-93G mineral mix. ^4^ AIN-93-VX vitamin mix. ^5^ BHT = Butylated hydroxytoluene. ^6^ Expressed as average ± standard deviation.

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
