# Peer review of "Characterization of a Coproduct from the Sea Cucumber Cucumaria frondosa and Its Effects on Visceral Adipocyte Size in Male Wistar Rats"

_marinedrugs, 2020, doi:10.3390/md18110530_

Round 1

Reviewer 1 Report

The authors have investigated the benefits of a sea cucumber coproduct on cardiometabolic health using a rat model. The results demonstrated that the coproduct showed smaller epididymal adipocyte size when 1.5% coproduct was added in the diet as compared to the control while no effect being seen for the lower and higher doses. The dose of 0.5% decreased epididymal fat mass and epididymal + retroperitoneal fat masses compared to the dose of 2.5%; however did not differ from the control. However, the body composition was not altered by the treatments as compared with the control and not different among the treatment groups either. The dose of 1.5% decreased blood concentration of VLDL+LDL-c compared to the dose of 0.5% while did not differ from the dose of 2.5%.  All other parameters measured were not different among the three doses and the control. Below are a few minor comments.  

  • Check the footnotes of table 7. The number for some notes were not matched with the number in the table. Also, suggest to remove the first footnote “n=10” as it is for the diet composition instead the treatment groups.
  • Fecal collection was performed for a period of 3 days by manual collection when the rats were kept in the same solid bottom cages. This method might have resulted in the contaminations of feces by food and other materials available such as beddings. Rodents also eat their own feces, resulting in the inaccuracy of fecal collection due to the partial reentering of feces into the digestive system. A proper way to do it is to use metabolic cages. Please discuss this.
  • The fecal samples were collected in three days and separated for the analysis of lipid content. In this case, a repeated measures ANOVA should be used for fecal lipid analysis for days 1, 2, and 3.
  • A few of analyses were done in singulate, please provide the reason(s) why not in duplicate at least.
  • For those variables that were not normally distributed, did they become normal after one-time data transformation as specified?
  • The authors selected the amount of the sea cucumber coproduct based on the no effect level, below the lowest effective level and largely above the lowest effective level. The results showed the opposite trend of effect, suggesting that the optimal dose could be lower than 0.5%. If there were any literature supports this notion, or at least further experiments are warranted to determine it.
  • In tables 1 and 2, data were presented as means ± standard error. In the section of Statistical Analysis, it was mentioned that all analyses were conducted in duplicate unless specified. Please provide clarify this.
  • In comparison with the control, neither of the treatments showed a significant effect on any one of the parameters measured. As such, it is not possible to make a conclusion that the sea cucumber coproduct is beneficial to cardiometabolic health. Please revise the Conclusion accordingly.

Author Response

Response to Reviewer #1

Manuscript ID: marinedrugs-927140

The authors have investigated the benefits of a sea cucumber coproduct on cardiometabolic health using a rat model. The results demonstrated that the coproduct showed smaller epididymal adipocyte size when 1.5% coproduct was added in the diet as compared to the control while no effect being seen for the lower and higher doses. The dose of 0.5% decreased epididymal fat mass and epididymal + retroperitoneal fat masses compared to the dose of 2.5%; however did not differ from the control. However, the body composition was not altered by the treatments as compared with the control and not different among the treatment groups either. The dose of 1.5% decreased blood concentration of VLDL+LDL-c compared to the dose of 0.5% while did not differ from the dose of 2.5%.  All other parameters measured were not different among the three doses and the control. Below are a few minor comments. 

  • Check the footnotes of table 7. The number for some notes were not matched with the number in the table. Also, suggest to remove the first footnote “n=10” as it is for the diet composition instead the treatment groups.

We thank the Reviewer for this comment. We have removed the first footnote of n=10 as suggested by the reviewer, and furthermore specified in tables 3 to 6 that n=10/diet. Errors in the footnote numbering have also been addressed. See lines in the revised manuscript 123, 147, 149 to 152, 154, 173, and 522 to 524.

  • Fecal collection was performed for a period of 3 days by manual collection when the rats were kept in the same solid bottom cages. This method might have resulted in the contaminations of feces by food and other materials available such as beddings. Rodents also eat their own feces, resulting in the inaccuracy of fecal collection due to the partial reentering of feces into the digestive system. A proper way to do it is to use metabolic cages. Please discuss this.

We thank the Reviewer for these relevant comments. We chose not to use metabolic cages and instead perform a manual collection since the use of metabolic cages could have induced a state of stress in the rats that, in turn, may have influenced other physiologic measurements that we consider of greater importance. We do acknowledge that the manually collected feces may have been contaminated with bedding and food. Although we were very meticulous when collecting the feces, we cannot formally exclude contamination. When adding the coprophagic nature of rats, we acknowledge the limitations of our assessment. In response to this comment, we have chosen to remove all discussion of fecal lipids from the main manuscript and supplemental materials.

  • The fecal samples were collected in three days and separated for the analysis of lipid content. In this case, a repeated measures ANOVA should be used for fecal lipid analysis for days 1, 2, & 3.

As stated in the previous comment, we have opted to simply remove the sections pertaining to fecal lipids.

  • A few of analyses were done in singulate, please provide the reason(s) why not in duplicate at least.

The HDL-C and the VLDL+LDL-C were both measured in singlicate due to the limited supply of serum obtained at the time of sacrifice.

  • For those variables that were not normally distributed, did they become normal after one-time data transformation as specified?

Yes, all the non-normally distributed variables became normally distributed after one transformation using a Shapiro-Wilk test. This has been specified in the statistical analysis section. See lines 658 to 659.

  • The authors selected the amount of the sea cucumber coproduct based on the no effect level, below the lowest effective level and largely above the lowest effective level. The results showed the opposite trend of effect, suggesting that the optimal dose could be lower than 0.5%. If there were any literature supports this notion, or at least further experiments are warranted to determine it.

Deciding on the dosages of sea cucumber coproduct to use was arduous due to several factors. Saponins are a wide class of molecules and their properties can vary greatly depending on their molecular structure. As pointed out in the manuscript, the coproducts contain their own unique saponins that are not found in the body wall. The reverse is also true. Due to the lack of studies specifically considering saponins from sea cucumber coproducts, we had to extrapolate data from other sources. Due to the potential toxic nature of saponins, we based our doses on toxicological studies in rats. As explained, the high dosage group served to determine if there were potential unfavorable effects at higher doses. However, it is important to keep in mind that these toxicological studies used saponins derived from plant sources, which may have different potency compared to sea cucumber saponins. Furthermore, the toxicological assessment studies were mainly based on hematological and histopathologic anomalies of the digestive tract as wells as determination of the no observed adverse effect level and the lowest observed adverse effect level. In our study, the main outcome (decrease in adipocyte size) was observed at the medium dose of 1.5%. As mentioned in the manuscript the dose of saponins in the 1.5% group was similar to the doses employed by Hu et al and Meng et al, who observed decreases in the mass of visceral and subcutaneous adipose tissue with said doses. However, in another study, Hu et al did observe more effects than we did in healthy rats with a lower dosage of saponins extracted from the body wall of C. frondosa. We attributed the fact that we observed fewer significant effects than Hu et al to the potential differences in saponin molecules as well as potential interactions between compounds in the coproduct. However, as the Reviewer aptly pointed out, we cannot exclude that the coproduct may be more effective at a lower dose. The study by Hu et al does provide some support to this notion. However, the results of our study alone do not provide enough information. More research is warranted to determine which doses are the most effective, and based on this information, lower doses should be considered. In light of this information, we have added a sentence pointing out the possibility that the coproduct may be more effective at lower doses and that this should be considered in future studies. See lines 314 to 315.

  • In tables 1 and 2, data were presented as means ± standard error. In the section of Statistical Analysis, it was mentioned that all analyses were conducted in duplicate unless specified. Please provide clarify this.

The data on chemical composition of the coproduct from the Sea Cucumber presented in Tables 1 and 2, as well as in Supplementary Table S1, are in fact presented as means ± standard deviation. The previous statement that means ± standard errors were reported was an error due to miscommunication between collaborators. Reporting standard deviation was more appropriate than reporting standard error since, for these values, we only have two individual analyses of the same sample for each variable. It has been corrected/added in the title of Table 1, Table 2 and Supplement Table S1. All other data are presented as means ± standard error as written in the Statistical Analysis section. The mention that all analyses were conducted in duplicate unless specified was necessary because analyses for fatty acids content (Table 2) were conducted in triplicates and for amino acids (Supplementary Table S1), in 4 replicates (or 3 replicates for tryptophane). It also has been corrected in the title of the Tables to clarify this issue. See lines 93, 118 to 119, and 642 to 644 as well as Supplementary Table S1.

  • In comparison with the control, neither of the treatments showed a significant effect on any one of the parameters measured. As such, it is not possible to make a conclusion that the sea cucumber coproduct is beneficial to cardiometabolic health. Please revise the Conclusion accordingly.

We thank the Reviewer for this comment. Indeed, only epididymal adipocyte size was significantly smaller in the 1.5% group compared to the control group. We were perhaps too bold in inferring a benefit regarding visceral adipose tissue weight due to how similar the values of the control group were to the values of the 2.5% group. In light of this comment, we determined that it would be best to put the emphasis on the observed decrease in adipocyte size. Alterations have been made to the conclusion, as well as to the discussion and abstract, and the observations regarding the masses of the adipose tissues are no longer included in these sections. The title of the article has also been altered to better reflect our main findings.

See lines 4, 26, 28, 188 to 190, 269 to 276, 284, 288 to 298, 315 to 317, 377 to 378, and 670 to 671.

Reviewer 2 Report

The manuscript by Ramalho and co-authors (927140), entitled:

“Characterisation of a Coproduct from the Sea Cucumber Cucumaria frondosa and its Effects on Cardiometabolic Health in Male Wistar Rats” provides very interesting and novel information on new potential sources of bioactive compounds from coproduct of sea cucumber that showed positive effects on several metabolic health parameters in male Wistar rats.

The MS is written in good scientific language. Introduction and discussion show the deep understanding the issue, scientific problems and a need of further research in this direction. Materials and methods describe experimental design, data collection and statistical analysis in details. Conclusions are clear as well as a strength and limitation of the work.

I have only one comment regarding Table 2. In my opinion, the amount of unidentified fatty acids seems too high, especially if to compare it with those of other FA groups, especially that all groups are present in the table. So, what class of these FAs may it be? Also, “total” would be better to replace with “a sum of omega-3 and omega-6”.

In my opinion, this paper should be accepted for publication to “Marine Drugs”.

Author Response

Response to Reviewer #2

Manuscript ID: marinedrugs-927140

“Characterisation of a Coproduct from the Sea Cucumber Cucumaria frondosa and its Effects on Cardiometabolic Health in Male Wistar Rats” provides very interesting and novel information on new potential sources of bioactive compounds from coproduct of sea cucumber that showed positive effects on several metabolic health parameters in male Wistar rats.

The MS is written in good scientific language. Introduction and discussion show the deep understanding the issue, scientific problems and a need of further research in this direction. Materials and methods describe experimental design, data collection and statistical analysis in details. Conclusions are clear as well as a strength and limitation of the work.

  • I have only one comment regarding Table 2. In my opinion, the amount of unidentified fatty acids seems too high, especially if to compare it with those of other FA groups, especially that all groups are present in the table. So, what class of these FAs may it be? Also, “total” would be better to replace with “a sum of omega-3 and omega-6”.

We thank the Reviewer for these comments and suggestions.

Precisions about unidentified fatty acids have been added to the results and discussion sections. The complete profile of fatty acid has been added in Supplementary Table S2. The 6 unidentified fatty acids are ≤C15, but from the data we have, we cannot determine if these unidentified fatty acids were saturated fatty acids with unusual structures (ex: branched chain) or unsaturated fatty acids.

See lines 107 to 114 and 214 to 220 as well as Supplementary Table S2.

Regarding the amount of PUFAs expressed in Table 2, the term “total” has been replaced with “Sum of Omega 3 & 6” of the PUFAs as suggested by the reviewer.

See Table 2 (line 119).

Reviewer 3 Report

The manuscript by Ramalho et al. " Characterization of a Coproduct from the Sea Cucumber Cucumaria frondosa and its Effects on Cardiometabolic Health in Male Wistar Rats" is a very well performed study aimed at identifying potential health benefit of using a byproduct from Cucumaria frondoza. The study is methodologically well done and provides indication that at low dose of food supplement made from byproducts of C. frondoza, animals (rats) demonstrate some reduction in adipose tissue and adipocyte size.

The authors rightfully state that the composition of byrpoduct is very complex and the effect of it on adipose tissue cannot be attributed to any one isolated group of molecules. Special discussion and attention is given to saponins that are known to possess strong biological activities that include immnomodulatory and hemolytic properties. Also, high arsenic content in C. frondoza is demonstrated and discussed from the point of view of its toxicity.
Probably, there is only one aspect that was not fully discussed in this paper, namely, strong seasonal variation in PUFA content in C. frondoza that is characteristic to many marine invertebrates. Probably, it would help to include that into discussion and provide additional references, for example,
https://dalspace.library.dal.ca/xmlui/handle/10222/73011
SEASONAL VARIATION OF LIPID AND CAROTENOID CONTENT OF CUCUMARIA FRONDOSA VISCERA OIL AND THE DEVELOPMENT OF METHODS FOR ITS QUALITY CONTROL

Also, it would be a good idea to mention a seminal contribution of A.A.Benson in the study of arsenolipid biochemistry when discussing arsenolipid occurrence and distribution within marine food chain.

Author Response

We thank the Reviewer for these suggestions. We also thank the Reviewer for suggestions of literature to assist in developing upon their suggestions. Indeed, season variations in lipid content should be considered. Additions have been made to the main manuscript that address these variations. See lines 221 to 236.

Regarding arsenic, further information regarding the accumulation of arsenic in marine photosynthetic lifeforms has been added, along with a referral to the works of A.A. Benson for more information on the subject. See lines 244 to 255.

Reviewer 4 Report

The paper provides detailed information concerning food supplement obtained from sea cucumber (Cucumaroa frondosa). This paper is submitted to the special issue "Cardio-Protective Properties of Marine Natural Products" however, this paper is only loosely tied with cardiovascular health as it is declared in the title.

My suggestions:

Table 1 and Table 7  the contents of columns could be left-aligned. Similarly, the rest of the tables could be aligned to the left in my opinion ot could be easier to read. One space could be introduced in Table 1 between categories mentioned in column 1.

I to could be probably easier to understand for the readers if in tables simply statistically significant difference between the tested group and control is presented.

Table 1.  Are values for Potassium, Sodium and  Phosphorus really right are SD values really 0?

Authors could provide some representative UPLC chromatograms obtained in amino acids analysis, they may include them in supplementary material, however, in maim text, they also could be interesting for some readers. Similarly, ICP-MS spectra could be presented in the main text or in the supplementary material

Were rats kept really individually in their cages?

The Authors presented a detailed description of the histopathologic assessment. Again it could be interesting for the reader to see some representative microscopic images, as well as pictures used for further analysis by the method described by Laforest at all. Some explanations indicating analysed points could be really interesting.

Author Response

Response to Reviewer #3

Manuscript ID: marinedrugs-927140

The paper provides detailed information concerning food supplement obtained from sea cucumber (Cucumaroa frondosa). This paper is submitted to the special issue "Cardio-Protective Properties of Marine Natural Products" however, this paper is only loosely tied with cardiovascular health as it is declared in the title.

My suggestions:

  • Table 1 and Table 7 the contents of columns could be left-aligned. Similarly, the rest of the tables could be aligned to the left in my opinion ot could be easier to read. One space could be introduced in Table 1 between categories mentioned in column 1.

We thank the Reviewer for these comments and suggestions. All the tables in the main manuscript have been reformatted to be aligned to the left to ease comprehension. Additionally, spaces have been added between the categories presented in Table 1. Furthermore, the footnote in Table 1 has been redone to be clearer and to be consistent with the other tables.

See tables 1 to 7 and line 94.

  • I to could be probably easier to understand for the readers if in tables simply statistically significant difference between the tested group and control is presented.

We thank the Reviewer for this suggestion. Indeed, the significant differences between experimental groups but not the control groups (namely the epididymal adipose tissue mass, the sum of the epididymal and retroperitoneal mass, and the VLDL+LDL-C)  presented in some tables could be misleading if not given careful attention. However, we respectfully suggest that in the results section, although there was no significant difference from the control regarding these variables, they provide information that should not be completely ignored, particularly regarding the mass of the adipose tissues. Although, the mass of these adipose tissues in the 1.5% group was significantly different from the 2.5% group and not the control, the similar values shared by the control and 2.5% groups suggest that if the power to detect significant differences was increased (i.e. a greater number of rats per group), a significant difference from the control group could potentially be observed. While in this study we cannot infer this, we do believe it is important to be considered for future research. In order to ease comprehension of the tables, in the cases where there were significant differences between experimental groups, but not the control group, an Asterix was placed next to the P value. This indication was also explained as footnotes to the corresponding tables and in the statistical analysis section.

See lines 150 to 151 and 665 to 666.

Furthermore, the title, the discussion, and the conclusion have been adjusted to put the emphasis on the observed decrease in adipocyte size. Much of the text pertaining to adipose tissue mass has been removed from these two sections.

See lines 4, 26, 28, 188 to 190, 269 to 276, 284, 288 to 298, 315 to 317, 377 to 378, and 670 to 671.

Table 1.  Are values for Potassium, Sodium and  Phosphorus really right are SD values really 0?

The values for potassium, sodium and phosphorus were the same for the two independent measurements, so the reported SD is 0.

  • Authors could provide some representative UPLC chromatograms obtained in amino acids analysis, they may include them in supplementary material, however, in maim text, they also could be interesting for some readers. Similarly, ICP-MS spectra could be presented in the main text or in the supplementary material

Representative UPLC chromatograms obtained in amino acid analysis have been added in Supplementary Figure S1. Representative GC chromatograms obtained in fatty acid analysis have been added in Supplementary Figure S2. As indicated in Materials and Methods, mineral analysis was performed by an external laboratory (Groupe Environnex, Longueil, Québec). This laboratory is accredited ISO 17025 by the Standards Council of Canada. Metals analysis by mass spectrometry in argon plasma (ICP-MS) in food and edible products (human and animal consumption) (ILCE-069) is within their scope of accreditation. They provided us with the results of mineral analysis, but not the ICP-MS spectra.

See Supplementary Figures S1 and S2 as well as lines 95 to 96 and 107 to 113

  • Were rats kept really individually in their cages?

Yes, the rats were kept individually in their cages. The main purpose for this measure was to accurately track food intake of each individual rat.

  • The Authors presented a detailed description of the histopathologic assessment. Again it could be interesting for the reader to see some representative microscopic images, as well as pictures used for further analysis by the method described by Laforest at all. Some explanations indicating analysed points could be really interesting.

The Authors thank the reviewer for this suggested. As suggested, representative examples of the microscopic images and the converted images were added to supplementary materials, with references to them in the main manuscript. See Supplementary Figures S3 and S4 as well as lines 628 to 629 and 631 to 632.

Round 2

Reviewer 1 Report

In Table S1, please provide in the footnote how amino acid score was calculated and the specify the reference pattern (for example, the reference pattern for 2-5 years old child, FAO/WHO, 2013) used for the calculation.

Author Response

In Table S1, please provide in the footnote how amino acid score was calculated and the specify the reference pattern (for example, the reference pattern for 2-5 years old child, FAO/WHO, 2013) used for the calculation.

The Authors thank the Reviewer for the request for clarification. As requested, a footnote has been added to Supplemental Table S1 with the additional information. The method used was obtained from the WHO/FAO/UN report from 2007. The requirement pattern numbers for adults can be found in table 49 of said document.

Reviewer 3 Report

Authors answered reviewer's questions

Author Response

Authors answered reviewer's questions.

The Authors thank the Reviewer for their time and consideration in reviewing this manuscript.

Reviewer 4 Report

The authors have fulfilled my requests and answered my questions, the paper may be accepted.

Author Response

The authors have fulfilled my requests and answered my questions, the paper may be accepted.

The Authors thank the Reviewer for their time and consideration in reviewing this manuscript.